# Beneficial Use of the Combination of Gemcitabine and Dacarbazine in Advanced Soft Tissue Sarcomas: Real-World Data

**DOI:** 10.3390/cancers16020267

**Published:** 2024-01-08

**Authors:** Ibon Gurruchaga Sotés, M. Carmen Gómez-Mateo, María Eugenia Ortega Izquierdo, Javier Martínez-Trufero

**Affiliations:** 1Department of Medical Oncology, Hospital Universitario de Navarra, 31008 Pamplona, Spain; 2Instituto de Investigación Sanitaria de Aragón, 50009 Zaragoza, Spain; meortegai@salud.aragon.es (M.E.O.I.); oncojmt@gmail.com (J.M.-T.); 3Department of Pathology, Hospital Universitario Miguel Servet, 50009 Zaragoza, Spain; mcgomezmateo@hotmail.com; 4Department of Medical Oncology, Hospital Universitario Miguel Servet, 50009 Zaragoza, Spain

**Keywords:** advanced soft tissue sarcoma, gemcitabine, dacarbazine, prognostic index, biomarker

## Abstract

**Simple Summary:**

Advanced soft tissue sarcomas (aSTSs) have scarce treatment options due to their low incidence, being considered by the World Health Organization as a rare disease; low investment by principal actors in trial designs; and high variability in treatment responses. The combination of gemcitabine and dacarbazine has been demonstrated to be effective in aSTSs in terms of progression-free survival and overall survival in three phase I–II clinical trials. Some pathological, clinical, and analytical variables have been analyzed as possible prognostic and predictive factors, providing a clue for clinicians to select the most optimal treatment in this setting.

**Abstract:**

Background: The combination of gemcitabine and dacarbazine has exhibited efficacy in terms of progression-free survival (PFS) and overall survival (OS) for aSTSs, albeit without robust confirmation from larger clinical trials. Methods: We conducted a retrospective study in a single institution involving aSTS patients treated with gemcitabine and dacarbazine. Results: 95 patients were assessed, pointing to a benefit in PFS of 3.5 months and an OS of 14.2 months. Patients with translocated histotypes had better PFS, while those with platelet–lymphocyte ratios (PLRs) surpassing a specific threshold or lower albumin levels had poorer overall survival. Conclusions: This study validates previous findings from three phase I–II trials, affirming the utility of this treatment approach in routine clinical practice.

## 1. Introduction

Soft tissue sarcomas (STS) are mesenchymal tumors that comprise more than 80 different entities, as defined by the last classification of the World Health Organization (WHO), based on histopathological, immunohistochemical (IHC), and molecular findings [1].

Sarcomas are considered rare diseases, and according to the RARECARE project, the incidence of STSs (excluding gastrointestinal stromal tumors) is 4–5/100,000 inhabitants per year [2], taking into account that their incidence, prevalence, and mortality are difficult to quantify. However, evidence suggests that actual incidence and prevalence are underestimated [3].

Similar to other tumors, in advanced STSs (aSTSs) the goal of treatment, in addition to prolonging overall survival (OS), should focus on controlling symptoms and improving quality of life. Unlike the management of localized STSs, which depends on the location of the primary tumor, advanced disease treatment comprehends other factors such as histological and molecular features. The standard first-line treatment typically involves anthracycline-based chemotherapy (CT), predominantly doxorubicin monotherapy [4]. Other studies explore the benefit of alternative treatment combinations with or without anthracyclines depending on the histology, such as weekly paclitaxel in angiosarcomas [5], NTRK inhibitors in tumors with a rearrangement of this gene [6,7,8], and doxorubicin combined with trabectedin in the case of leiomyosarcomas (LMSs) [9], as well as in combination with ifosfamide in the event that the aim of the treatment is achieving volumetric responses in some STS subtypes [4].

However, considering the second line of treatment, international guidelines advocate for histotype-driven treatments [10,11]. Various therapeutic options, such as trabectedin [12], eribulin, or different tyrosine kinase inhibitors [13,14,15,16], have been evaluated through international multi-institutional trials and approved by regulatory institutions.

Beyond approved drugs, other tested CT regimens have shown benefits, even in OS. For instance, the combination of gemcitabine with docetaxel or dacarbazine. A phase III clinical trial, GeDDiS, compared the combination of docetaxel and doxorubicin as a first-line treatment for metastatic disease, resulting in similar PFS and 2-year mortality in both arms; that is why the authors recommended maintaining the standard with doxorubicin [17]. In second and subsequent lines, one trial carried out by RG Maki et al. compared this combination to gemcitabine in monotherapy with objective response rates (ORRs) of 16% and 8%, median progression-free survival (mPFS) of 6.2 and 3 months, and median OS (mOS) of 17.9 and 11.5 months, respectively, though this was not statistically significant (*p* = 0.052) [18]. In TAXOGEM, a similar trial but limited to LMS, the authors also pointed out that the addition of docetaxel to gemcitabine hardly provides substantial clinical benefit [19].

However, despite these findings, we have a potentially effective combination, gemcitabine together with dacarbazine, evaluated in three phase I–II clinical trials [20,21,22], with disease control rates (DCR) of 49% vs. 25% when compared to gemcitabine alone, mPFS of 4.2 vs. 2 months, and mOS of 16.8 vs. 8.2 months, respectively [22]. In addition to this OS benefit, toxicity was manageable, highlighting hematologic adverse events, mostly reversible and solved before the beginning of the next cycle. In the subgroup analysis, both in the univariate and in the multivariate analysis, the study showed that the LMS subtype predicted a better response to combination therapy, with a mPFS and mOS of 4.9 and 18.3 months, respectively, compared to non-LMS histologies, with mPFS and mOS of 2.1 and 7.8 months, respectively.

While this combination has not gained widespread acceptance as a viable second-line option, except in some European countries like Spain, given the limited treatment options for aSTSs, there is a genuine need to maximize its potential. It is crucial to identify the patients who could benefit from this approach in real-life scenarios.

Consequently, we conducted a uni-institutional real-world analysis collecting data from 95 patients with aSTSs treated with gemcitabine and dacarbazine. The aim was to validate the outcomes reported in the trial led by X. García-Del-Muro et al. [22], wherein a median PFS and OS of 3.5 and 14 months, respectively, were achieved. Additionally, we aimed to identify potential clinical and biological factors beyond the histological subtype that could offer prognostic information.

## 2. Materials and Methods

### 2.1. Patients and Objectives

A retrospective observational study was conducted on patients aged over 16 with a confirmed histological diagnosis of STS (patients with locally advanced stage non-candidates for “curative” surgery and those with metastatic setting) treated using the gemcitabine and dacarbazine combination for at least 2 cycles between July 2007 and December 2020 at the Medical Oncology Department of Miguel Servet University Hospital, within normal clinical practice.

The main objective of the study was to determine the clinical benefit of the combination scheme, assessed via DCR (the sum of patients with partial response or radiological stable disease) and PFS. Secondary objectives included OS, the analysis of clinicopathological characteristics, and the identification of reported toxicities and biological prognostic factors related to both the tumor and the patient, correlating them with disease outcomes.

#### 2.1.1. Treatment

Gemcitabine and dacarbazine treatment administration dosage was based on the comparative phase II clinical trial carried out by X. García-Del-Muro et al. [22]. Gemcitabine was administered at a fixed-dose rate of 10 mg/m^2^/min over a 180 min intravenous infusion followed by dacarbazine 500 mg/m^2^ intravenously over 20 min every 14 days. Dose adjustments were made based on patient toxicity and clinical discretion, following standard clinical practice (see Appendix A, Table A1). Toxicity grades were assessed according to the Common Terminology Criteria for Adverse Events (CTCAE) version 5 [23].

#### 2.1.2. Prognostic Markers

Various potential prognostic and predictive markers from hemograms and biochemistry, with normal values from the hospital’s central laboratory, were analyzed. These included neutrophils (normal value 2000–7500 per mL), lymphocytes (normal value 1500–4000 per mL), platelets (normal value 150,000–400,000 per mL), and erythrocyte distribution width (RDW) (11–15%) as well as albumin concentration (3.4–5.4 g/dL) and lactate dehydrogenase (140–280 IU/L).

Derived indices from these biomarkers were calculated, such as the neutrophil–lymphocyte ratio (NLR), with a cutoff point of 2.5 based on a bibliographic review [24,25], and the platelet–lymphocyte ratio (PLR), with a cutoff point of 190 [26].

Another indirect measure of treatment effectiveness analyzed in this study is the growth modulation index (GMI), calculated as the ratio of the time to progression of the study treatment divided by the time to progression of the previous line, GMI > 1.33 being the cut-off point for defining a treatment as effective [27].

### 2.2. Statistical Method

Binomial variables were collected and expressed by their frequency distribution (frequencies and percentages), while quantitative variables are described as the mean, median, and standard deviation. Survival analyses are shown through the representation of the Kaplan–Meier estimator. Comparisons between qualitative variables were carried out using the Fisher Exact Test or Chi-square. In the case of quantitative and qualitative variables, Mann–Whitney U or Kruskal–Wallis tests were used.

In the study, both PFS and OS were analyzed, the starting point being the administration of the first cycle of gemcitabine–dacarbazine. The final moment for PFS is progression, according to the RECIST criteria (version 1.1) [28], and/or clinical symptomatic progression. First, the survival function and the cumulative risk function (Kaplan–Meier limit product estimator) were analyzed in a purely descriptive manner, contrasting whether the risk function was different depending on the presence of certain factors (bivariate study), for which the Kaplan–Meier limit product estimator was used, using the Mantel–Haenszel (log-rank) contrast by pairs of groups. Afterward, the Cox proportional hazard model (better known as Cox regression) was used to estimate a model that analyzes the influence of the covariates as independent prognosis factors for both PFS and OS. To carry out the contrasts in the quantitative covariates, they were grouped into intervals. The *p*-values reported were two-sided, and the significance level was 0.05. SPSS statistics version 25.0 was used for the statistical analysis of the study.

## 3. Results

Between July 2007 and December 2020, a total of 95 patients treated with the combination of gemcitabine and dacarbazine in unresectable locally advanced or metastatic settings were collected.

### 3.1. Baseline Characteristics of the Population

Gender distribution was nearly equal, with 50.5% women (*n* = 48) and 49.5% men (*n* = 47). The mean age at diagnosis was 59 years (ranging from 19 to 84 years).

Regarding the histological subtypes, the most frequent were LMS (27.4%, *n* = 26), undifferentiated pleomorphic sarcoma (UPS) (22%, *n* = 21), and, subsequently, with percentages around 6–7%, not otherwise specified (NOS) sarcoma (8%, *n* = 8), liposarcoma (LPS) (6%, *n* = 6), malignant peripheral nerve sheath tumor (MPNST) (6%, *n* = 6), and synovial sarcoma (5%, *n* = 5).

Primary tumor locations varied, with the trunk or extremities accounting for 43%, followed by gynecological origin (16%), retroperitoneum (16%), viscera (11%), and less than 10% in other locations.

In relation to the above, principal metastatic locations were multisystemic involvement (34.7% | *n* = 35), exclusive pulmonary involvement (29.5% | *n* = 28), and only unresectable locoregional involvement (25.3% | *n* = 24).

Finally, highlighting the treatments received prior to the administration of gemcitabine–dacarbazine, the most widely used scheme was doxorubicin in monotherapy or in combination with other agents (85.3%, *n* = 81). This included patients treated in the first line of metastatic disease and those who previously received it in an adjuvant setting but had a non-surgical relapse within a year. The median number of prior treatment lines was two, with 67.4% of patients having received between two and three lines previously. Refer to Table 1 for a summary of all baseline characteristics.

### 3.2. Effectiveness of the Combination

#### 3.2.1. Response Rates

The treatment was administered for an average of eight cycles (ranging from 1 to 104 cycles) with a median of 4.5 cycles. Out of the total patients treated, there were 2 complete responses (CRs) (2.1%), 19 partial responses (PRs) (20%), 29 patients with stable disease (SD) (30.5%), 44 patients with progressive disease (PD) as the best response (46.3%), and 1 patient who died before the first radiological evaluation.

We present in Figure 1 an example demonstrating the effectiveness of gemcitabine and dacarbazine in a patient with advanced retroperitoneal LMS who underwent treatment for more than 30 cycles, maintaining a PR.

#### 3.2.2. Survival Data

The mPFS was 3.5 months (95% CI 1.85–5.15 months) (see Figure 2a). The median OS for the entire study population was 14.2 months (95% CI 10.23–18.1 months) (see Figure 2b), taking the date of administration of the first cycle of gemcitabine–dacarbazine as a reference point. In the OS analysis, six patients were censored due to loss of follow-up.

An indirect indicator of treatment efficacy, the growth modulation index (GMI), yielded a median of 1.26 in our study, with 32.6% of patients exhibiting a GMI > 1.3. In this comparison, we had a 32% rate of loss data.

### 3.3. Combination Safety Profile

The reported adverse events documented in clinical history were present in 75.8% of patients, with a low percentage of serious side effects (grades 3–4). The most relevant toxicity was hematological, the most common being anemia, which was present in 64.2% (*n* = 61) of patients, but only one suffered from anemia of grades 3–4. The second most frequent toxicity was thrombocytopenia, present in 27.4% of patients (*n* = 26), but with a drop to 3% for grades 3–4. Finally, neutropenia affected 8.4% (*n* = 8) of the patients, of whom only five (5.3%) presented grade 3–4 neutropenia. No febrile neutropenia or toxic deaths were reported, as seen in Table 2.

7.4% of patients (*n* = 7) needed some kind of dose reduction, and 8.4% of patients (*n* = 8) had their treatment definitively stopped due to the deterioration of their performance status.

### 3.4. Prognostic and Predictive Factors of Response

We conducted univariate and multivariate analyses on potential prognostic variables (Table A2 and Table A3). Univariate analysis for PFS revealed several statistically significant prognostic markers, including leiomyosarcoma histology, molecular complexity, translocated subtypes, single pulmonary metastases, and the presence of PR or CR. However, in a multivariate COX regression model, the presence of translocated histological subtype (*p* = 0.036) or objective response (*p* < 0.000) emerged as independent prognostic markers associated with greater PFS (Table 3).

In the univariate analysis for OS, several factors emerged as statistically significant prognostic markers, including leiomyosarcoma, sarcoma NOS, total lymphocyte count at baseline, albumin concentration at baseline, RDW at the end of treatment, objective response to gemcitabine–dacarbazine, L-sarcomas (encompassing leiomyosarcoma and liposarcoma), and PLR before treatment initiation. Upon conducting multivariate COX regression analysis, certain variables exhibited independent prognostic value. Notably, a baseline total lymphocyte count < 1000 cells/mm^2^ amplified the risk of death by 83.8% compared to counts above this threshold. Additionally, a one-unit increase in baseline albumin corresponded to a 47.7% reduction in the risk of death. Conversely, a one-unit increase in RDW at the end of treatment escalated the risk of death by 17.8%. Patients showcasing an objective response to the treatment experienced a 3.522-fold decrease in the risk of death compared to those with SD or PD. Finally, patients with a PLR exceeding 190 before treatment initiation faced a 76.6% higher risk of death compared to others (Table 4).

According to guidelines, second-line and subsequent treatments should be based, among other criteria, on histological subtype. Despite this, and although in the univariate analysis, presenting an LMS showed statistically significant better rates of PFS and OS, this was not the case after the multivariate analysis. Figure 3 reflects the PFS curves (Figure 3a) and OS (Figure 3b) according to the histological subtype.

## 4. Discussion

There have been multiple clinical trials that have evaluated different therapeutic schemes beyond the first line in unresectable locally advanced and/or metastatic STS, but nevertheless, few have shown a clear benefit in PFS and even less in OS. Regarding the use of gemcitabine–dacarbazine, three phase I–II clinical trials have proved such a benefit [20,21,22].

To the best of our knowledge, this is the first retrospective study to date that validates with real-world data the benefit in survival rates of the use of gemcitabine–dacarbazine in non-selected patients with aSTSs.

Our study’s demographic characteristics closely resemble X. Garcia-Del-Muro’s trial, with a slightly older median age (59-year-old vs. 49 to 51-year-old), a similar distribution by sex and the location of the primary tumor, and a similar distribution by histology, with LMS, UPS, and sarcoma NOS more predominant in our population (27%, 22%, and 8%, respectively) compared to the trial population, in which the prevalent histologies were LMS, LPS, UPS, and sarcoma NOS (29%, 18%, 17%, and 17%, respectively). All are shown in Table 5 [22].

When examining survival rates, our study demonstrated a PFS of 3.5 months and an OS of 14.2 months, slightly inferior to Garcia-Del-Muro’s study, with an OS of 16.8 months. The results of our study are easily understandable in the context of non-selected real-world patients. The aim of this study is not only to validate the results of García-Del-Muro’s study but also to evaluate this underused combination, probably due to limited knowledge in the medical community. We should highlight that we have few clinical trials with an active arm, thus the direct comparison of chemotherapeutic agents is minimal. Taking this into account, assuming a risk of inconsistency in comparing prospective and retrospective data while validating a real-world clinical trial seems quite similar and even better than the results obtained in other trials that have led to the approval of several chemotherapeutic agents. Examples are trabectedin [12,29], pazopanib [14], or eribulin [30], whose survival data are shown in Table 6. Among these drugs, it should be noted that eribulin has only been studied in LPS and LMS, with a PFS benefit of less than 3 months and an OS of 13.5 months. The combination of gemcitabine with docetaxel, evaluated in several trials, such as the TAXOGEM clinical trial [19] and the trial by RG Maki et al. [18], failed to demonstrate the benefit of adding docetaxel to gemcitabine. These studies show a PFS of less than 5 months for gemcitabine monotherapy and an OS of 11.5 months. For trabectedin, a single-arm study showed a PFS of 3.5 months and an OS of 13.9 months [12,29], and another placebo-controlled study showed a benefit for pazopanib in terms of PFS and OS of 4.6 and 12.5 months, respectively [14].

An indirect criterion of treatment effectiveness that has also been analyzed within the study is the GMI. This is an index defined at the end of the 20th century by Von Hoff, making an intra-patient comparison between the CT scheme that precedes gemcitabine–dacarbazine versus the latter, where each patient acts as their own control. After this value was analyzed by the French Sarcoma Group and identified as a surrogate for OS [31], several studies have attempted to extrapolate it to the use of trabectedin, obtaining values below the cut-off point [32,33]. In our study, the median GMI value exceeds the previous results, obtaining a value of 1.26. Therefore, even though we could not reach the cut-off value, we can consider that these data improve what has been described so far. In any case, this is the first time that these data have been analyzed in patients treated with such a scheme. It has to be said that we were not able to analyze the entire study population since in up to 32% of the patients, the PFS of the scheme prior to the use of gemcitabine–dacarbazine was not calculated, since they were patients who had relapsed during or close to adjuvant treatment or for whom survival data could not be collected.

In terms of toxicity, notably improved results in hematological toxicity were observed compared to previous clinical trials involving this combination. There was a 20% reduction in anemia rates of any grade and a 15% and 60% reduction in thrombocytopenia and neutropenia of any grade, respectively, the most striking being a 10-fold reduction in the risk of grade 3–4 neutropenia. These outcomes can be attributed, in part, to the utilization of granulocyte colony-stimulating factors, blood transfusions, and dose reductions. It should be noted that neither previous studies nor ours reported any treatment discontinuations due to intolerable toxicity. In our study, dose reductions were applied in 7.4% of the population without an impact on PFS. Comparatively, when compared with previously mentioned drugs, except for pazopanib and eribulin, others exhibited higher rates of hematological toxicity.

Finally, within the study, we analyzed different variables in order to see their prognostic influence on this specific population. Two factors emerged as having independent prognostic value in the multivariate analysis. Firstly, we noticed that histological subtypes with translocation [34] presented an HR of 1.872 (*p* = 0.036), predicting improved PFS, a variable already described as a predictor of response to trabectedin [35,36,37]. However, there is no evidence in the literature so far about this influence with the use of gemcitabine-based combinations. Secondly, logically, those patients who presented objective responses showed better PFS, with an HR of 2715 (*p* < 0.0001).

Peripheral blood indices have been extensively studied for their prognostic roles across various cancer types, including STSs. In our study, considering OS, the basal lymphocyte count, PLR, and basal albumin concentration were independently related to OS. Previous studies in the field of STSs have also identified other indices, such as the NLR or the Glasgow Prognostic Score, as having prognostic value [38]. One retrospective study found that elevated levels of both NLR and PLR predicted a higher risk of death, with an HR for NLR of 1.698 (*p* < 0.001) and for PLR an HR of 1.346 (*p* < 0.001) [26]. In our study, while the NLR value did not reach significance, PLR exhibited a significant HR of 1.766 (*p* = 0.038). Additionally, our study highlighted that lymphopenia or hypoalbuminemia are predictors of lower PFS, variables commonly used to calculate the Prognostic Nutritional Index [39].

Our study has obvious limitations due to its design, as it is a single-center retrospective study with a limited number of patients and different STS histotypes. Despite that, it must be taken into account that our number of patients is in line with most phase II trials in STSs, since it is very difficult to carry out studies in low-prevalence tumors.

## 5. Conclusions

In this pioneering study utilizing real-world data on the efficacy of the gemcitabine–dacarbazine scheme across a larger patient cohort compared to the experimental arm in Garcia-Del-Muro et al.’s study [22], we substantiated the therapeutic benefits of this regimen. In fact, our findings showed superior survival rates compared to more extended therapeutic schemes. Furthermore, our study identified specific subpopulations that could obtain more benefit from this scheme, such as those patients with translocated subtype tumors, among others. Additionally, we observed that several baseline analytical parameters—such as albumin levels, lymphocyte counts, or pre-treatment PLR—hold predictive value for improved survival outcomes. These insights emphasize the potential for targeted and more effective treatment strategies in STSs.

## Figures and Tables

**Figure 1 cancers-16-00267-f001:**
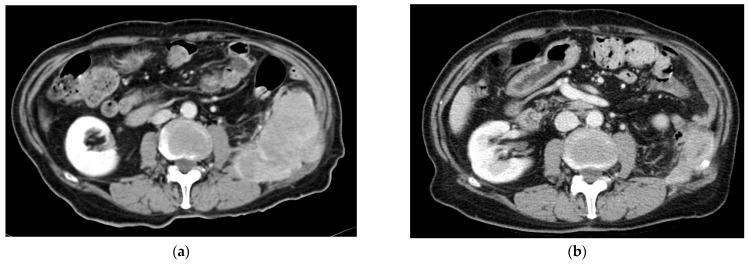
Patient with an advanced retroperitoneal leiomyosarcoma treated with gemcitabine and dacarbazine: (**a**) Computed tomography (CT) before treatment initiation (January 2017). (**b**) CT showing a maintained partial response after 1.5 years under the same treatment (October 2018).

**Figure 2 cancers-16-00267-f002:**
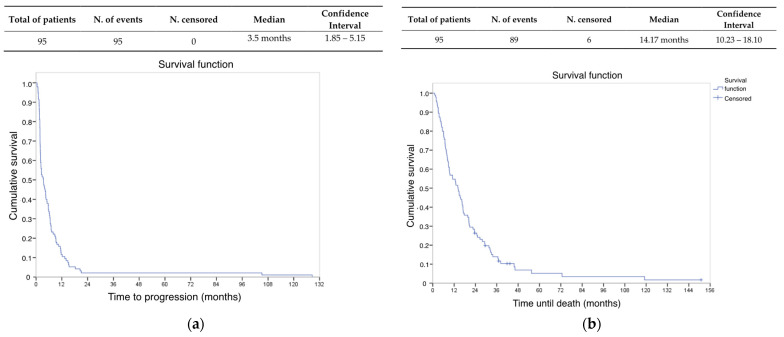
Kaplan–Meier curves for gemcitabine–dacarbazine according to (**a**) progression-free survival and (**b**) overall survival.

**Figure 3 cancers-16-00267-f003:**
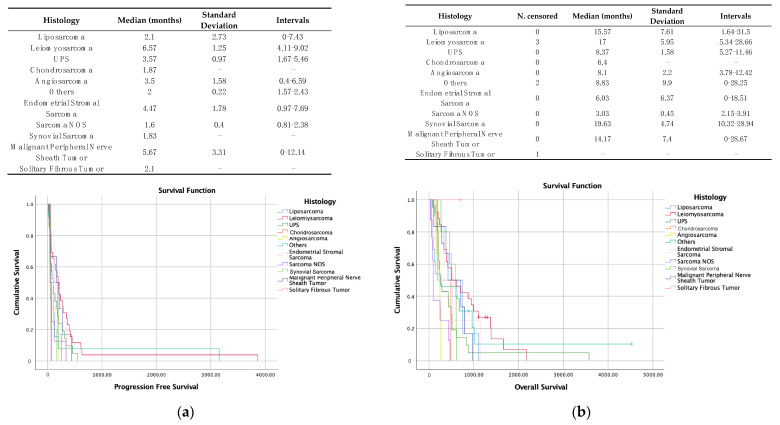
Descriptive analysis and Kaplan–Meier curves for gemcitabine–dacarbazine subclassified according to histology. (**a**) Progression-free survival. (**b**) Overall survival.

**Table 1 cancers-16-00267-t001:** Baseline characteristics of the population.

Baseline Characteristics	*n* = 95
Mean age	At diagnosis: 55 years old At unresectable disease: 59 years old
Sex	Men 49.5% (*n* = 47) Women 50.5% (*n* = 48)
Histological subtypes	LMS 27.4% (*n* = 26) UPS 22% (*n* = 21) NOS sarcoma 8% (*n* = 8) LPS 6% (*n* = 6) SS 4% (*n* = 5) Others 30.5% (*n* = 29)
Complex karyotype	78.7% (*n* = 74)
Primary tumor location	Trunk and limbs 43% (*n* = 41) Retroperitoneum 15.8% (*n* = 15) Gynecologic 15.8% (*n* = 15) Viscera 11.6% (*n* = 11) Others 13.7% (*n* = 13)
Stage of palliative disease	Multisystemic 24.7% (*n* = 33) Lung 29.5% (*n* = 28) Locoregional non-surgical 26% (*n* = 24) Hepatic 2% (*n* = 2) Other localizations 5% (*n* = 5)
Previously used chemotherapy agents	Anthracycline combination or monotherapy 85% (*n* = 83) Ifosfamide 7.3% (*n* = 7) Combination of taxane 4.2% (*n* = 4) Trabectedine 1% (*n* = 1)
Treatment after progression to GD	Trabectedine 36.8% (*n* = 35) TKI 16.8% (*n* = 16) Others 2% None 29.5% (*n* = 28)

LMS: leiomyosarcoma. UPS: undifferentiated pleomorphic sarcoma. NOS sarcoma: not otherwise specified sarcoma. LPS: liposarcoma. SS: synovial sarcoma. GD: gemcitabine–dacarbazine. TKI: tyrosine kinase inhibitor.

**Table 2 cancers-16-00267-t002:** Toxicity profile of the combination.

Toxicity	Any Grade	Grades 3–4
Anemia	64.2% (*n* = 61)	1% (*n* = 1)
Thrombopenia	27.4% (*n* = 26)	3.2% (*n* = 3)
Neutropenia	8.4% (*n* = 8)	5.3% (*n* = 5)
Febrile Neutropenia	0%	0%
Total	75.8% (*n* = 72)	9.5% (*n* = 9)

**Table 3 cancers-16-00267-t003:** COX regression for PFS.

Covariate	HR	CI (HR)	Significance
Translocated histological subtype	1.87	1.74–4.89	0.036
ORR	2.71	1.60–4.6	0.000

HR: hazard ratio. CI: confidence interval. ORR: overall response rate.

**Table 4 cancers-16-00267-t004:** COX regression for OS.

Covariate	HR	CI (HR)	Significance
Lymphocyte at beginning of treatment	1.84	1.08–3.13	0.026
Albumin at beginning of treatment	0.52	0.33–0.81	0.004
ORR	3.52	1.99–6.23	0.000
PLR previous to start of treatment	1.77	1.03–3.02	0.0038

HR: hazard ratio. CI: confidence interval. ORR: overall response rate. PLR: platelet–lymphocyte ratio.

**Table 5 cancers-16-00267-t005:** Comparative table of the principal variables of reference studies with GD.

Variable	Our Study	JM Buesa [20]	R. Losa [21]	X. Garcial-Del-Muro [22]
N° of patients	95	22	26	59 vs. 54
PFS	3.5 months	At 6 months 29% free of progression	At 6 months 28% free of progression	4.2 months (vs. 2 months with gem.)
OS (months)	14.2	-	8.63 months	16.8 months (vs. 8.2 months with gem.)
ORR	22%	26%	4%	
SD	30.5%	31.6%	47.8%	ORR + SD 49% (vs. 25% with gem.)
Toxicities	Anemia 64.2% (G3-4: 1%) Thrombopenia 27.4% (G3-4: 3%) Neutropenia 8.4% (G3-4: 5.3%)	Anemia 83% (G3-4: 5%) Thrombopenia 11% (G3-4: 5%) Neutropenia 100% (G3-4: 44%)	Anemia 92% (G3-4: 33%) Thrombopenia 58% (G3-4: 12%) Neutropenia 73% (G3-4: 46%)	Anemia 82% (G3-4: 4%) Thrombopenia 40% (G3-4: 6%) Neutropenia 76% (G3-4: 48%)

PFS: progression-free survival. OS: overall survival. ORR: overall response rate. SD: stable disease. G: grade. Gem: gemcitabine.

**Table 6 cancers-16-00267-t006:** Comparative table of the principal variables of reference studies with other chemotherapies.

Variable	Nº Patients	PFS (Months)	OS (Months)	Response	Toxicities
Our study	95	3.5	14.2	CR 2.1% PR 20% SD 30.5%	Anemia 64.2% (G3-4: 1%) Thrombopenia 27.4% (G3-4: 3%) Neutropenia 8.4% (G3-4: 5.3%)
Gemcitabine [18,19]	22–49	3–4.7	11.5	PR 8–19% ORR 8%	Anemia 85% (G3-4: 1–13%) Neutropenia 59% (G3-4: 21–28%) Thrombopenia 60% (G3-4: 8–35%)
Gemcitabine–Docetaxel [18,19]	24–73	5.5–6.2	17.9	PR 16–24% ORR 16%	Anemia 94% (G3-4: 7–10%) Neutropenia 41% (G3-4: 10–16%) Thrombopenia 62% (G3-4: 18–40%)
Eribulin [30]	228	2.6	13.5	PR 4% SD 56%	Anemia 30% (G3-4: 16%) Neutropenia 43% (G3-4: 35%) Thrombopenia 6% (G3-4 < 1%)
Pazopanib [14]	246	4.6	12.5	PR 6% SD 67% PD 23%	-
Trabectedin [29]	270	3.5	13.9	PR 8% SD 26%	Anemia 97% (G3-4: 8%) Thrombopenia 53% (G3-4: 11%) Neutropenia 74% (G3-4: 47%)

CR: complete response. PR: partial response. SD: stable disease. ORR: overall response rate. G: grade.

## Data Availability

The datasets used and/or analyzed during the current study are available from the corresponding author upon reasonable request.

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
