# Peer review of "Beneficial Use of the Combination of Gemcitabine and Dacarbazine in Advanced Soft Tissue Sarcomas: Real-World Data"

_cancers, 2024, doi:10.3390/cancers16020267_

Round 1
Reviewer 1 Report
Comments and Suggestions for Authors
Mamuscript entitled "Beneficial use of the combination of gemcitabine and dacarbazine in advanced soft tissue sarcomas"
1. Given that sarcoma is a heterogeneous group, the authors should clearly described the histological type and the response of different type of sarcoma.
2. The authors should provide representative radiographic and pathology images for selected cases to make this study sound.
3. The laboratory data should be provided in more detail.
Comments on the Quality of English LanguageAcceptable.
Reviewer 2 Report
Comments and Suggestions for Authors
The authors try to address an interesting question, that is studying real-world administration of Dacarbazine-Gemcitabine in patients with metastatic soft tissue sarcomas. They measure patient- and tumor-related features and try to correlate these to PFS and OS. They also try to compare the observed PFS and OS to the data of a clinical trial from X. García-Del-Muro et al. that employed a similar regimen for patients with STS.
There are several issues that I see with this manuscript, which however I believe can be improved and lead to its publication.
First, the statistical methods and presentation of the results need to be improved.
In the Statistical Method section there are several comments describing the different types of tests used to assess differences in categorical or continuous variables. However, the results from these tests are not presented in the text or table. I would suggest to add at least a p-value in the tables whenever a comparison of a variable is made between groups and to add a notation at the bottom of the table to indicate what test was used to obtain that p-value.
When describing how PFS and OS are measured, it needs to be explicitly described how cases were censored. Also censoring data should be reported on the KM curves. Also, which software did the authors use for statistical analysis? That must be disclosed.
Also, please describe in more details why the patients had "advanced" disease. This is briefly mentioned in one of the tables but must be more clear.
Another issue is regarding the Cox analysis. Only data for the multivariate analysis is presented. I recommend to make a table for the univariate analysis and a table for the multivariate results. Also, usually the same set of variables are tested against PFS and OS, but here the authors test a different set of variables. Why is that? There should be a justification or some sort of rationale as to why a given variable was included for testing for PFS and not for OS and vice-versa.
Also, in the description of toxicity, the authors refer to grades of adverse reaction. Which grading system did they use? This should be clearly stated in the methods. Also why is "Lost by the system" an adverse reaction?
Last but not least, the authors try to make a lot of comparisons with several clinical trials. It must be clearly noted in the discussion and conclusions that from a statistical standpoint no conclusions can be drawn by comparing a retrospective study to a prospective study like a clinical trial.
Comments on the Quality of English LanguageThere are several grammatical errors throughout the manuscript. It is overall understandable even though some passages are harder to read.
Round 2
Reviewer 1 Report
Comments and Suggestions for Authors
The revision is acceptable in the present form.
Comments on the Quality of English LanguageAcceptable
Reviewer 2 Report
Comments and Suggestions for Authors
The authors have addressed my concerns.